# Study of the Antioxidant and Anti-Inflammatory Properties of the Biological Extracts of *Psophocarpus tetragonolobus* Using Two Extraction Methods

**DOI:** 10.3390/molecules26154435

**Published:** 2021-07-22

**Authors:** Hussein Bassal, Akram Hijazi, Hussein Farhan, Christine Trabolsi, Bouchra Sayed Ahmad, Alia Khalil, Marc Maresca, Fawaz El Omar

**Affiliations:** 1Doctoral School of Science and Technology, Research Platform for Environmental Science (PRASE), Lebanese University, Beirut P.O. Box 6573, Lebanon; hussein.bassal@ul.edu.lb (H.B.); Christinetrabolsi9@gmail.com (C.T.); Bouchra.ahmad27@gmail.com (B.S.A.); alia.khalil.be@gmail.com (A.K.); fomar@ul.edu.lb (F.E.O.); 2Laboratory of Cancer Biology and Molecular Immunology, Faculty of Sciences, Lebanese University, Beirut P.O. Box 6573, Lebanon; 3Biotechnology Department, College of Science, University of Baghdad, Baghdad 10070, Iraq; husseinfarhan95@gmail.com; 4CNRS, Centrale Marseille, iSm2, Aix Marseille University, 13397 Marseille, France

**Keywords:** winged bean, nitric oxide synthase, cyclooxygenase-2, ultrasound-assisted extraction, maceration, interleukin-6

## Abstract

*Psophocarpus tetragonolobus* has long been used in traditional medicine and cuisine. In this study, *Psophocarpus tetragonolobus* extracts were isolated by maceration and ultrasound-assisted extraction and were evaluated for their antioxidant and anti-inflammatory effects in lipopolysaccharide (LPS)-stimulated RAW264.7 macrophages. The obtained results show that both extracts (maceration and ultrasound) were rich in bioactive molecules and exerted substantial antioxidant and anti-inflammatory effects. The *P. tetragonolobus* extracts’ treatment in LPS-stimulated RAW264.7 macrophages resulted in a significant downregulation of the expressions of tumor necrosis factor-α (TNF-α), interleukin-6 (IL-6), and IL-1β mRNA. In addition, the *P. tetragonolobus* extracts’ treatment attenuated inducible nitric oxide synthase (iNOS) and cyclooxygenase-2 (COX-2) protein expression. Our observations indicate that there is no significant difference between the two studied extracts of *P. tetragonolobus* in terms of biological properties (specifically, antioxidant and anti-inflammatory effects. Regardless of the extraction method, *P. tetragonolobus* could be used for treating diseases related to oxidative stress and inflammatory reactions.

## 1. Introduction

Inflammation is one of the first immune and physiological defense processes in response to extrinsic and intrinsic damage such as that from microorganisms, tissue injury, chemical agents, irradiation, and improper immunological responses [1]. However, uncontrolled and excessive inflammatory responses may contribute to the initiation and progression of several inflammatory-mediated diseases such as atherosclerosis, rheumatoid arthritis, cardiovascular diseases, and cancer [2].

The essential part of the inflammatory response is played by innate immune cells such as macrophages, neutrophils, and mononuclear phagocytes that contribute to the early sensing and identification of dangerous signals through their surface-expressed pattern recognition receptors (PRRs; such as Toll-like receptors). These signals may include pathogen-associated molecular patterns (PAMPs) (e.g., lipopolysaccharides LPS in Gram-negative bacteria, microbial nucleic acids, lipoproteins, and carbohydrates) and/or signals derived from host cellular damage such as danger-associated molecular patterns (DAMP) and environmental stressors [3].

Following activation, macrophages cause enhanced production of several pro-inflammatory cytokines and mediators such as interleukin-1β (IL-1β), interleukin-6 (IL-6), tumor necrosis factor alpha (TNF-α), nitric oxide (NO), and prostaglandins (PGs) [4,5].

The inducible forms of nitric oxide synthase (iNOS) and cyclooxygenase-2 (COX-2) enzymes, which are responsible for the secretion of NO and PGs, respectively, are shown to participate in the pathogenesis of several diseases including cancer, neurodegenerative diseases, and atherosclerosis [6,7]. COX-2 overexpression has been reported to enhance the metastatic potential and invasiveness of cancer cells and correlates with poor patient prognosis [8]. On the other hand, increased iNOS activity and expression have been positively correlated with the degree of malignancy in gastric cancer, squamous cell carcinoma, hepatocellular carcinoma, melanoma, and leukemia [9].

In addition to producing inflammatory cytokines, prostaglandins, and chemokines, polymorphonuclear neutrophils (PMNs) are also a source of reactive oxygen species (ROS). ROS are defined as partially reduced metabolites of oxygen with strong oxidizing capacities. At the site of inflammation, ROS such as superoxide (O^2−^) can rapidly combine with nitric oxide (NO) to form reactive nitrogen species (RNS) such as peroxynitrite [10]. ROS and RNS act as both signaling molecules and mediators of inflammation. At physiological concentrations, they serve complex signaling functions such as cell growth, adhesion, differentiation, senescence, and apoptosis. However, an enhanced ROS generation causes endothelial dysfunction, tissue injury, and oxidative stress. The latter is a harmful process that can damage cellular constituents (protein, lipid, and DNA) and has been associated with the development of various critical pathologies, including cancer, arthritis, atherosclerosis, diabetes, and cardiovascular and neurodegenerative diseases [11]. Therefore, the inhibition of inflammatory response may contribute to the prevention of inflammation-associated diseases.

Although several anti-inflammatory drugs are available, many of them can cause detrimental side effects on the gastrointestinal tract and kidney after long-term use [12]. Thus, there has been a focus on studies aiming at the discovery of new anti-inflammatory therapeutic agents.

Plants are well known for their medicinal value, primarily related to their phytochemical component content, mainly phenolic compounds, flavonoids, alkaloids, and tannins. The mechanism of action by which flavonoids and phenolic compounds function as antioxidant and anti-inflammatory factors is thought to lie in their free radical scavenging activities and inhibition of pro-inflammatory enzymes such as COX and lipoxyoxygenase (LOX). The biological activities of these natural compounds have been reported in several in vitro studies as well as in numerous preclinical studies [13].

*Psophocarpus tetragonolobus* (L.) DC, commonly known as winged bean, asparagus pea, or Goa bean, is a self-pollinated tropical legume that belongs to the family Fabaceae.

This plant is also known for its nutritional value containing vitamins (Vitamins A and C), minerals (calcium and iron), erucic acid, polyunsaturated fatty acids, and protein (30–45%, namely lectins) [14,15]. Historically, the beans have been commonly used in traditional medicine for many years: the leaves are boiled to make a decoction or infusion used as a lotion to cure smallpox or as a wash for inflammation and suppurating sores, and raw leaves and seeds are eaten to cure skin sores (such as boils and ulcers), can help in strengthening the body’s immune system against many infections, and are known for aiding in the prevention of cancer, diabetes, and asthma [16].

Apart from being an edible plant, the species is reported to have anti-bacterial [17], anti-proliferative [18], and fungicidal activities [19] and is now gaining the attention of researchers. So far, neither analytical nor biological studies have been performed on the Lebanese winged bean.

Extraction method plays a critical role in the study of medicinal plants [20]. The most common and oldest method used is dynamic maceration. This technique is based on the diffusion or transfer of the molecules of interest from a solid phase to an aqueous phase, where organic solvents are used according to their polarity. However, the major inconvenience of this extraction method is that it can cause a loss of bioactive compounds due to the use of high temperatures and long extraction times and generally gives a low extraction yield [21].

Therefore, significant advances have been made, and modern techniques, which reduce time and the need for energy-consuming procedures and increase extraction efficiency, have been used, including ultrasound-assisted extraction (UAE). The process utilizes the energy of sound waves to cause a pressure variation in the liquid extractor, generating a so-called acoustic cavitation effect, which creates shear forces that disrupt cell walls and facilitate the transfer of metabolites to the solvent [22,23].

Our purpose is to study the antioxidant and anti-inflammatory properties of the biological extracts of Lebanese Psophocarpus tetragonolobus using two extraction methods, maceration and ultrasound.

## 2. Materials and Methods

### 2.1. Plant Collection and Powder Preparation

Fresh leaves of winged beans were collected in the region of Harouf, southern Lebanon, at 480 m altitude during spring. After harvesting, the plant was carefully cleaned with distilled water, cut into small pieces, and dried in the shade at room temperature, away from sunlight. After this period, the dried leaves were crushed and ground to a homogeneous fine powder using a POLYMIX (PX-MFC 90 D) (Avantor company, Radnor, PA, USA) grind mill and then kept in the dark at room temperature until use in different studies.

### 2.2. Apparatus and Chemicals

All of the used chemicals were of analytical grade. Methanol, sodium hydroxide, ethyl acetate, and dichloromethane were purchased from BDH England. Aluminum chloride, ferrous sulfate heptahydrate (FeSO_4_•7H_2_O), and silica gel were purchased from Merck (Darmstadt, Germany). Sodium carbonate and hydrogen peroxide were purchased from Unichem (Mumbai, India). Ascorbic acid, gallic acid, rutin, Folin–Ciocalteau reagent, EDTA, ferrozine, and 1,1-diphenyl-2-picrylhydrazyl (DPPH) were purchased from Sigma Aldrich (St. Louis, MO, USA). PBS was purchased from Gibco (Amarilo, TX, USA). Samples were weighed using an analytical and numerical balance (Mettler Toledo, Columbus OH, USA). The dried leaves were ground using a POLYMIX (PX-MFC 90 D) (Avantor company, Radnor, PA, USA) grind mill. The absorbance values of the solutions were measured using a VWR UV-6300PC double beam spectrophotometer, and the extracts were concentrated using a HEIDOLPH rotavapor apparatus (Heidolph Instruments GmbH & Co. KG, Schwabach, Germany).

### 2.3. Extraction Methods (Maceration and Ultrasound)

Powdered leaves (20 g) were deposited into a flask with 200 mL of the selected solvent (30% distilled water and 70% methanol), and two extractions methods were performed: maceration and ultrasound application. For maceration, the plant material was subjected to constant stirring for one week at room temperature. Then, the mixture was centrifuged (3000 rpm, 15 min) and filtered using filter paper. For ultrasound, the process was similar to that described above except that instead of soaking for one week: the mixture of plant solvent material was subjected to two 30 min periods of ultrasonic radiation (25 kHz) in an ultrasonic bath (TI-H-5, Elma Schmidbauer GmbH, Singen, Germany) using the same solvent system. Finally, both extracts were concentrated in a rotary evaporator at 40 °C under reduced pressure and dried under vacuum for 24 h at 30 °C.

### 2.4. Phytochemical Screening

To determine the chemical composition of the different extracts of *P. tetragonolobus*, qualitative tests were performed to detect the presence of primary and secondary metabolites as shown in Table 1. These tests are useful to estimate biological activities that might be due to the presence of secondary metabolites in this plant [24].

**Test for alkaloids (Dragendorff’s, Mayer’s, and Wagner’s tests):** One gram of dried plant was added with 10 mL of 1M-HCl and ultrasonicated for 15 min at 30 °C. The mixture was filtered, and 3 mL of filtrate was treated with a few drops of either Dragendorff’s reagent or Mayer’s reagent or Wagner’s reagent. Orange, red, creamy white, or reddish-brown precipitate indicated the presence of alkaloids [25].

**Test for resins (turbidity test):** Ten milliliters of distilled water was added to 1 g of dried plants and ultrasonicated for 15 min at 30 °C. The mixture was filtered. Occurrence of turbidity showed the presence of resins [26].

**Test for flavonoids:** Two milliliters of 2.0% NaOH mixture was mixed with aqueous plant crude extract; concentrated yellow color was produced, which became colorless when we added 2 drops of diluted acid to the mixture. This result showed the presence of flavonoids [27].

**Test for saponins:** A total of 0.2 g of extract was shaken with 5 mL of distilled water in a test tube. Frothing that persisted on warming was taken as evidence for the presence of saponins [28].

**Test for quinones:** To 1 mL of extract, 1 mL of concentrated sulfuric acid was added. Formation of red color indicated the presence of quinones [27].

**Test for phenolic compounds (ferric chloride test):** One gram of dried plant was added with 10 mL of ethanol and ultrasonicated for 15 min at 30 °C. The mixture was filtered, and 2 mL of filtrate was added with 5 mL of distilled water. The filtrate was treated with a few drops of 5%-FeCl_3_. Dark green color indicated the presence of phenolic compounds [29]. 

**Test for tannins (ferric chloride test):** One gram of dried plant was added with 10 mL of distilled water and ultrasonicated for 15 min at 80 °C. The mixture was filtered, and the filtrate was cooled down. Two milliliters of filtrate were treated with a few drops of 0.1% FeCl_3_. The presence of tannins was indicated by the brownish-green or blue-black coloration [30].

**Test for terpenoids:** One gram of dried plants was added with 10 mL of chloroform and ultrasonicated for 15 min at 30 °C. The mixture was filtered. Three milliliters of H_2_SO_4_ was added carefully to 5 mL of filtrate along the side of the test tube. Reddish-brown color at the interface of the two liquids characterized the presence of terpenoids [31].

**Test for carbohydrates:** The presence of carbohydrates was confirmed when 2 mL of extract was treated with 1 mL of Molisch’s reagent and a few drops of concentrated sulfuric acid, which resulted in the formation of purple or reddish color [27].

**Test for reducing sugars:** A total of 0.2 g of extract was shaken with distilled water and filtered. The filtrate was boiled with drops of Fehling’s solution A and B for two minutes. An orange precipitate on boiling with the Fehling’s solution indicated the presence of reducing sugars [28]. 

**Test for steroids:** Ten milliliters of ethanol was added to 1 g of dried plants and ultrasonicated for 15 min at 30 °C. The mixture was filtered, and the filtrate was evaporated to dryness. Two milliliters of chloroform was added to 100 mg of the crude extract. The mixture was added with 1 mL of glacial acetic acid, followed by careful addition of 1 mL H_2_SO_4_ along the side of the test tube. Greenish color indicated the presence of steroids [30]. 

**Test for cardiac glycosides** (Keller–Killani Test): Ten milliliters of ethanol was added to 1 g of dried plants and ultrasonicated for 15 min at 30 °C. The filtrate was evaporated to dryness. Two milliliters of glacial acetic acid and 2 drops of 2% FeCl_3_ were added to 100 mg of the crude extract. The mixture was added with 1 mL H_2_SO_4_ along the side of the test tube. Brown ring at the interface indicated the presence of cardiac glycosides [31].

**Test for anthraquinones:** A total of 0.2 g of extract was shaken with 4 mL of benzene. The mixture was filtered, and 2 mL of 10% ammonia solution was added to the filtrate. The mixture was shaken and the presence of pink, red, or violet color in the ammonical (lower) phase indicated the presence of free anthraquinones [31].

**Test for anthracyanins:** To 1 mL of extract, 1 mL sodium hydroxide was added and heated for 5 min at 100 °C. Formation of bluish-green color indicated the presence of anthocyanin [27].

### 2.5. Cell Culture

RAW 264.7 cells are a macrophage-like cell line transformed by Abelson leukemia virus and derived from BALB/c mice. RAW 264.7 cells were obtained from the American Type Culture Collection (ATCC, Rockville, MD, USA) and were used in our study. The cells were maintained in DMEM medium—High Glucose (Sigma, New Delhi, India) supplemented with 10% FBS and 1% penicillin-streptomycin (Sigma, New Delhi, India) at 37 °C in a humidified incubator under the pressure of 5% CO_2_. The macrophages were seeded in 12-well plates (1 × 10^5^ cells/well) using fresh medium. After pre-incubation for 24 h, plates were co-treated with LPS at 10 ng/mL and 2 different concentrations of the extracts (200 μg/mL and 300 μg/mL) in DMEM for an additional 24 h.

### 2.6. RNA Extraction and Quantitative Real-Time PCR

Extraction of total RNA was done using TRIzol reagent (Sigma, New Delhi, India) according to manufacturer’s instructions. RNA was quantified by determining the absorbance at 260 nm using a Nanodrop 2000 spectrophotometer (Thermo Scientific, MA, USA). One microgram of each sample was used for the production of cDNA using iScript Reverse Transcription Supermix (Bio-Rad, Hercules, CA, USA). Quantitative real-time PCR was performed using CFX96 PCR detection system (Bio-Rad, Hercules, CA, USA) apparatus and iTaq universal SYBR Green supermix (Bio-Rad, Hercules, CA, USA). The sequences of the primers used are listed in Table 2. The conditions used for qPCR were 95 °C for 3 min in the initial activation step, followed by 40 cycles consisting of 2 steps: 95 °C for 30 s, and 60 °C for 1 min. The relative quantity of mRNA was determined by Livak’s method [32] using 18 s rRNA as a reference gene and presented relative to cells incubated with LPS only.

### 2.7. Western Blot

A total of 5 × 10^5^ cells/well for each treatment were pelleted and lysed in RIPA lysis buffer supplemented with protease inhibitor. Cell lysates were centrifuged at 12,000× *g* for 15 min at 4 °C, and protein concentration was measured using the Bradford method (Bio-Rad). Proteins were separated on SDS-PAGE in reducing conditions and transferred to nitrocellulose membrane (Bio-Rad). After saturation with 5% milk, the membrane was incubated overnight at 4 °C with the appropriate primary antibody at the indicated dilutions—anti-COX-2 (Abcam—diluted 1:3000) and anti- iNOS (Abcam—diluted 1:2000)—washed, incubated with HRP-conjugated secondary antibody for 1 h at room temperature, and revealed using the ECL substrate (Bio-Rad). Monoclonal anti-GAPDH antibody (Abcam—diluted 1:10,000) was used for protein loading control. Densitometry analysis was performed using ImageJ.

### 2.8. DPPH Radical Scavenging Assay

The scavenging activity was measured using 1,1-diphenyl-2-picrylhydrazyl radical (DPPH). Increasing concentrations of extracts (100, 200, 300, 400, and 500 µg/mL) were tested. One milliliter from each solution was mixed with 1 mL of 0.5 mM DPPH and stored at room temperature in the dark for 30 min. The absorbance of the resulting solutions was measured at 517 nm against a blank using a Gene Quant 1300 UV-Vis spectrophotometer. The percentage of scavenging activity was calculated as
$$Scavenging\;activity=\frac{1-\left(A1-A2\right)}{A0}\times 100$$
where A0 is the absorbance of the control, A1 is the absorbance of the sample, and A2 is the absorbance of blank that contained sample without DPPH radical. The scavenging activity of each fraction was expressed as the IC_50_ value, the concentration required to scavenge 50% of DPPH.

### 2.9. Statistical Analysis

Data were expressed as mean ± standard error of the mean (SEM). For two groups’ comparison, unpaired t-test was used. Differences between multiple groups were evaluated using one-way or two-way ANOVA as appropriate, followed by Bonferroni post hoc test for multiple comparisons. The significance level was set according to *p*-value: *p* < 0.05 (*), *p* < 0.01 (**), and *p* < 0.001(***). All calculations were performed with GraphPad Prism 5.01.

## 3. Results

### 3.1. Phytochemical Screening Antioxidant Effect

Phytochemical screening of *P. tetragonolobus* crude extracts indicated the presence of some important bioactive components, which are listed in Table 3. Both maceration and ultrasound crude extracts showed the presence of alkaloids, tannins, resins, phenols, terpenoids, flavonoids, carbohydrates, diterpenes, proteins and amino acids, lignin, fixed oils, and lipids. In addition, the ultrasound crude extract exhibited the presence of sterols and steroids, and the maceration crude extract showed the presence of saponins.

### 3.2. Antioxidant Effect

The antioxidant activity of the crude extracts from both extraction methods was evaluated using a DPPH free radical scavenging assay. Our obtained results demonstrate that the two extracts displayed significant antioxidant activities and scavenging effects on DPPH radical with IC_50_ of 178.4 ± 1.5 and 157.5 ± 1.8 µg/mL for maceration and ultrasound extract, respectively (Figure 1).

### 3.3. In Vitro Anti-Inflammatory Activity

#### 3.3.1. Effect of *P. tetragonolobus* Crude Extracts on LPS-Induced Expression of TNF-α, IL-6, and IL-1β mRNA

Cells were co-treated with LPS (10 ng/mL) and each tested extract (200 μg/mL; M: maceration; U: ultrasound) for 24 h. mRNA was extracted and analyzed by qRT-PCR using primers targeting the indicated genes. Relative mRNA expression levels were normalized to 18s rRNA and expressed as fold change over LPS. Results were expressed as mean ± SEM of three independent experiments. Differences between multiple groups were evaluated using one-way ANOVA followed by Bonferroni’s Multiple Comparison Test (** *p* < 0.01, *** *p* < 0.001).

Having shown the antioxidant activity of the extract, quantitative real-time PCR was used to assess the anti-inflammatory effect of the winged bean crude extracts on RAW 264.7 cells after induction of inflammation by LPS (10 ng/mL). The two extracts were used at a concentration of 200 µg/mL (70% ethanol). The TNF-α, IL-6, and IL-1β mRNA levels were determined relative to cells treated only with LPS (Figure 2). Interestingly, the two extracts significantly reduced the LPS-induced upregulation of IL-1β and IL-6 mRNA expression (Figure 2A,B). Moreover, maceration and ultrasound crude extracts downregulated TNF-α mRNA levels by approximately 85.4 ± 9.4% and 87.0 ± 6.8%, respectively, at 200 µg/mL (Figure 2C).

#### 3.3.2. Effects of *P. tetragonolobus* Crude Extracts on LPS-Induced Expression of iNOS and COX-2 Protein Levels

Cells were co-treated with LPS (10 ng/mL) and each tested extract (200 μg/mL; M: maceration; U: ultrasound) for 24 h. Total protein extracts were made and subjected to Western blot analysis using the indicated antibodies. Representative Western blots were shown. Expression levels of COX-2 proteins (B) were quantified by densitometry and normalized to GAPDH and expressed as fold change over LPS. Results were shown as mean ± SEM of three independent experiments. Statistics were calculated using one-way ANOVA, followed by Bonferroni’s Multiple Comparison Test *** *p* < 0.001).

Western blot analysis was carried out to assess the protein levels of the pro-inflammatory mediators iNOS and COX-2, which were expressed upon inducing the macrophage-like cells with LPS. This was done after treatment with each *P. tetragonolobus* crude extract (200 µg/mL). As shown in Figure 3, LPS treatment induced the expression of both iNOS and COX-2 at the protein level. Interestingly, the two extracts were able to significantly reduce iNOS expression (by 100%). Similarly, a suppressive effect was exerted by maceration and ultrasound crude extracts on COX-2 protein levels (94.2 ± 1.3% and 94.4 ± 1.1% respectively, Figure 3B).

## 4. Discussion

In the present study, using two extraction methods (maceration and ultrasound-assisted extraction), we aimed to screen the chemical content of winged bean extracts and characterize their anti-inflammatory and antioxidant capacities. In terms of antioxidant activity, extracts obtained by the two extraction methods displayed similar activity.

Indeed, although the DPPH free radical scavenging effect of the extract obtained with the ultrasound-assisted extraction is slightly lower than that obtained with dynamic maceration (IC_50_ of 157.5 ± 1.8 versus 178.4 ± 1.5 µg/mL), our results did not highlight any remarkable differences in the biological properties of the two studied extracts. This could be explained by the fact that the phytochemical screening of both extracts was similar except for sterols and steroids, found only in the ultrasound-assisted extract, and saponins, found only in the maceration extract.

In terms of anti-inflammatory effect, our in vitro results showed that both extracts (maceration and ultrasound), at a concentration of 300 µg/mL, were efficient in suppressing IL1-β, TNF-α, and IL-6 mRNA expression, as well as inhibiting COX-2 and iNOS protein expression. Expectedly, in addition to phenols and flavonoids [33], phytosterols [34] and saponins [35] were shown to display antioxidant as well as anti-inflammatory activity, which justifies the similarity of biological effect between the two extracts.

## 5. Conclusions

Extraction is an important step involved in the discovery of bioactive components in medicinal plants. Two extraction methods, maceration and ultrasound, have been used to extract polyphenolic compounds from plant materials. Biological activities of plant extracts showed that both extracts (maceration and ultrasound) were rich in bioactive molecules and exerted substantial antioxidant and anti-inflammatory effects. The *P. tetragonolobus* extracts’ treatment in LPS-stimulated RAW264.7 macrophages resulted in a significant downregulation of the expressions of TNF-α, IL-6, and IL-1β mRNA. In addition, the *P. tetragonolobus* extracts’ treatment attenuated iNOS and COX-2 protein expression. The present study shows that winged bean could be a potential natural source of antioxidants and anti-inflammatory compounds. It could have greater importance as a therapeutic agent in preventing or slowing oxidative stress and inflammation-related disorders. Therefore, *Psophocarpus tetragonolobus* is a promising source of bioactive compounds that presents great potential as functional ingredients in the cosmetic industry. It can be promoted as a raw material in the cosmetic and/or cosmeceutical sectors due to its low price and abundance. In addition, this plant species has long been used for cooking ingredients and traditional medicines, which will help the cosmetic industry to decrease the cost of research and of studying its toxicity. Further investigations, using different geographical origins or different cultivation conditions to increase bioactive compounds in different plant parts of this species, are needed.

## Figures and Tables

**Figure 1 molecules-26-04435-f001:**
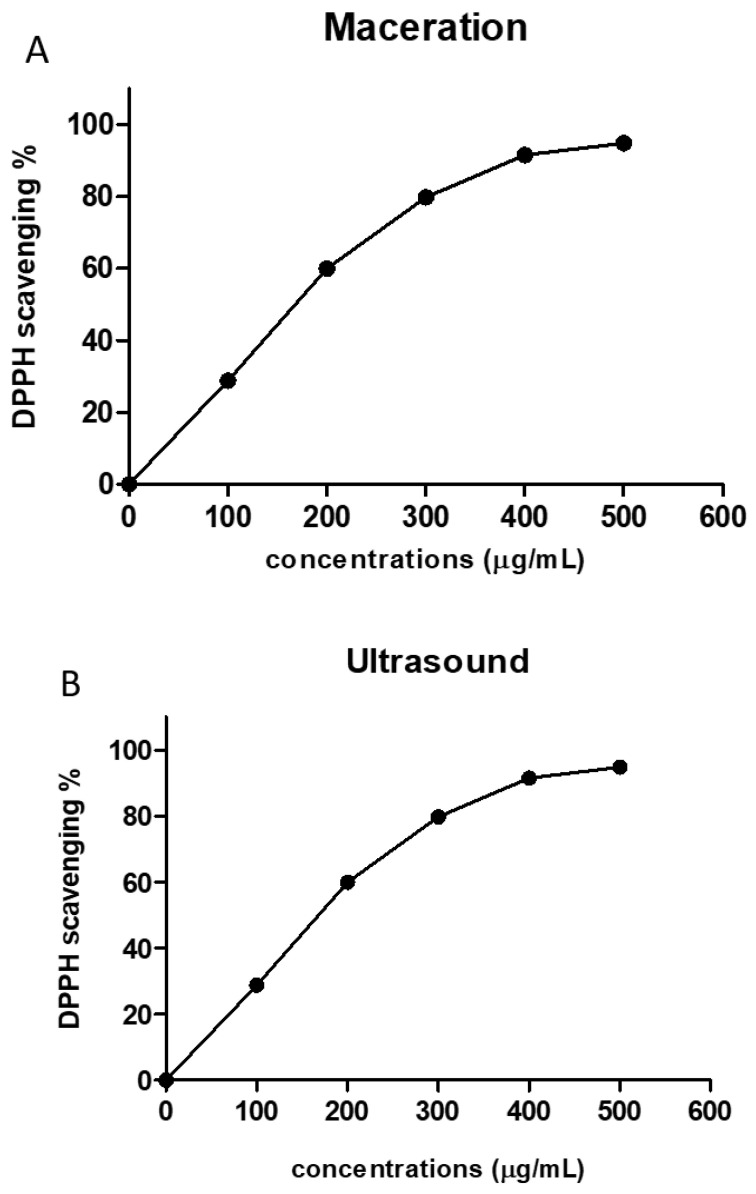
Antioxidant activities of *P. tetragonolobus* extracts according to the concentration of extract. DPPH scavenging activity of *P. tetragonolobus* extracts at different concentrations using Maceration (**A**) and ultrasound-assisted extraction (**B**) techniques. Each curve has been plotted using the means ± SD (*n* = 3).

**Figure 2 molecules-26-04435-f002:**
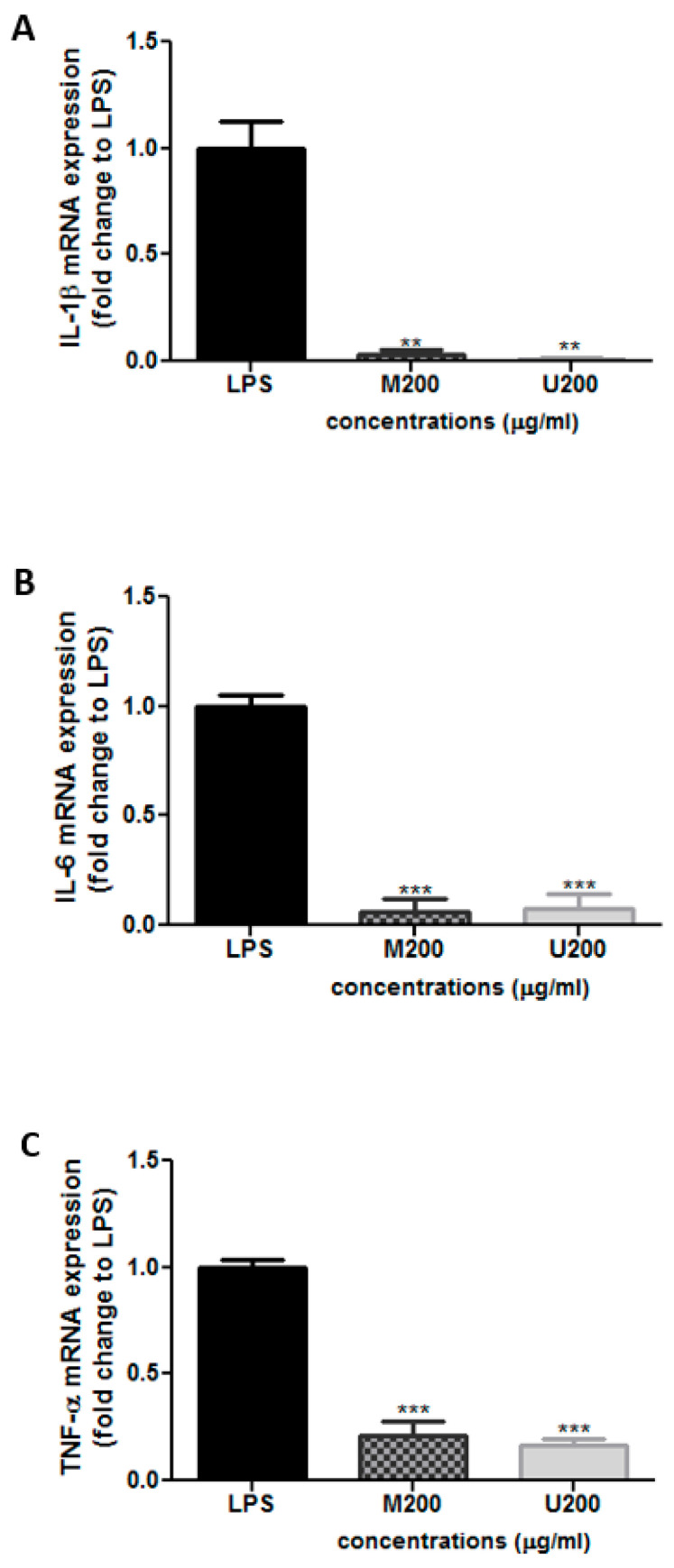
Relative mRNA levels of (**A**) IL-1β, (**B**) IL-6, and (**C**) TNF-α in LPS-treated RAW 264.7 cells by quantitative real-time PCR. (** *p* < 0.01, *** *p* < 0.001).

**Figure 3 molecules-26-04435-f003:**
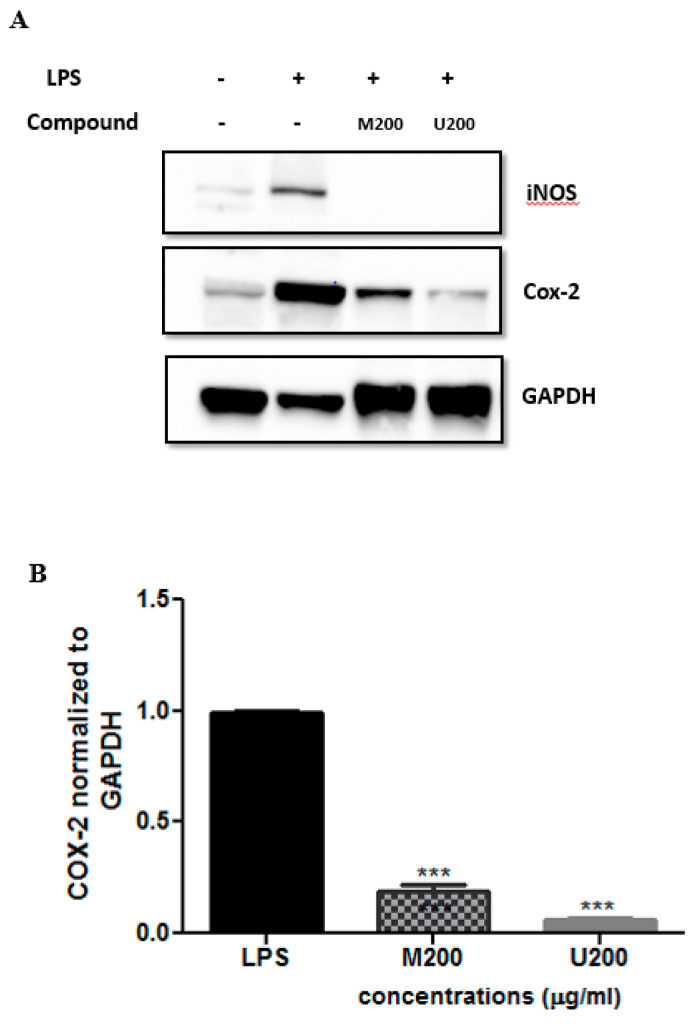
Protein expression levels of COX-2 and iNOS in LPS-treated RAW 264.7 cells assessed by Western blot. (**A**) Cells were co-treated with LPS (10 ng/ml) and each tested extracts (200 μg/mL; M: maceration; U: ultrasound) for 24 h. Total protein extracts were made and subjected to western blot analysis using the indicated antibodies. Representative western blots are shown. Expression levels of COX-2 proteins (**B**) were quantified by densitometry and normalized to GAPDH, and expressed as fold change over LPS. Results are shown as mean ± SEM of three independent experiments. Statistics were performed using one-way ANOVA, followed by Bonferroni’s Multiple Comparison Test. (*** *p* < 0.001).

**Table 1 molecules-26-04435-t001:** Detection of primary and secondary metabolites in leaves of *P. tetragonolobus*.

Metabolites	Added Reagent	Expected Result
Alkaloids	Dragendorff’s reagent	Red or orange precipitate
Resins	Acetone + water	Turbidity
Saponins	Agitation	Formation of foam
Quinones	Hydrochloric acid (HCl) concentrated	Red coloration
Phenols	FeCl_3_ (1%) + K_3_(Fe(CN)_6_) (1%)	Green-blue coloration
Flavonoids	Potassium Hydroxide (KOH) (50%)	Yellow coloration
Flavanones	H_2_SO_4_ concentrated	Bluish-red coloration
Carbohydrates	α-naphtol + H_2_SO_4_	Purple ring
Diterpenes	Copper acetate	Green coloration
Reducing sugars	Fehling’s solution (A + B)	Brownish-red precipitate
Sterols and steroids	Chloroform + H_2_SO_4_ concentrated	Red color (surface) + fluorescence Greenish-yellow
Cardiac glycosides	Glacial acetic acid + FeCl_3_ (5%) + H_2_SO_4_ conc	Ring
Anthraquinones	HCl concentrated (10%) + chloroform + Ammonia (10%)	Pink coloration
Proteins and amino acids	Ninhydrin 0.25%	Blue coloration
Lignins	Safranine	Pink coloration
Phlabotannins	HCl (1%)	Blue coloration
Anthocyanins	Sodium Hydroxide (NaOH) (10%)	Blue coloration
Fixed oils and fats	Spot Test	Oil stain

**Table 2 molecules-26-04435-t002:** List of primers used for the quantitative polymerase chain reaction (qPCR).

Gene	Forward Sequence (5′–3′)	Reverse Sequence (5′–3′)
18s rRNA	GCAATTATTCCCCATGAACG	GGCCTCACTAAACCATCCAA
TNF-α	GTAGCCCACGTCGTAGCAAACCAC	GGTACAACCCATCGGCTGGCAC
IL-6	CCTCTCTGCAAGAGACTTCCATCCA	TCCTCTGTGAAGTCTCCTCTCCGG
IL-1β	TGGACCTTCCAGGATGAGGACA	GTTCATCTCGGAGCCTGTAGTG

**Table 3 molecules-26-04435-t003:** Phytochemical screening of *P.tetragonolobus* extracts with two extraction methods.

Metabolite	Maceration	Ultrasound
Alkaloids	+	+
Tannins	+	+
Resins	+	+
Saponins	+	−
Phenols	+	+
Flavonoids	+	+
Flavanones	−	−
Reducing sugars	−	−
Quinones	−	−
Sterols and steroids	−	+
Cardiac glycosides	−	−
Diterpenes	+	+
Anthraquinones	−	−
Proteins and amino acids	+	+
Lignins	+	+
Phlabotannins	−	−
Anthocyanins	−	−
Fixed oils and fats	+	+

(−): absent; (+): present.

## Data Availability

The study did not report any data.

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
