# Peer review of "Study of the Antioxidant and Anti-Inflammatory Properties of the Biological Extracts of Psophocarpus tetragonolobus Using Two Extraction Methods"

_molecules, 2021, doi:10.3390/molecules26154435_

Round 1

Reviewer 1 Report

In this work, authors studied the effects of maceration and ultrasound assisted extraction on chemicals composition, antioxidant and anti-inflammatory activities of Psophocarpus tetragonolobus. As maceration and ultrasound assisted extraction are two widely used extraction methods, I failed to understand the novelty and relevance of this work. In addition, concerning the chemical composition, authors only performed screening assays to verify the presence/absence of several classes of compounds, and in my opinion, a detailed characterization of the extract with respective quantification is essential to better understand its biological effects and to establish relationships between the chemical composition and biological activity.

DPPH is a widely used assay to evaluate the antioxidant capacity; however, it is a synthetic radical and does not provide an approximate information about what might happen in the human organism. Thus, the work quality should have been improved if authors had studied the antioxidant activity against reactive species with biological significance.

Conclusion should be more concise and should highlight the main findings of the work.

Therefore, in the present form the work has not sufficient scientific impact to be published in Molecules.

Author Response

Reviewer 1 :

In this work, authors studied the effects of maceration and ultrasound assisted extraction on chemicals composition, antioxidant and anti-inflammatory activities of Psophocarpus tetragonolobus. As maceration and ultrasound assisted extraction are two widely used extraction methods, I failed to understand the novelty and relevance of this work. In addition, concerning the chemical composition, authors only performed screening assays to verify the presence/absence of several classes of compounds, and in my opinion, a detailed characterization of the extract with respective quantification is essential to better understand its biological effects and to establish relationships between the chemical composition and biological activity.

Thank you for this comment. The novelty of the study resides in the fact that is the first study of the antioxydant and anti-inflammatory properties of the biological extracts of Lebanese Psophocarpus tetragonolobus using two extraction methods maceration and ultrasound. Regarding the chemical analysis asked by Reviewer 1, we are really sorry but we will not be able to perform it, although it will interesting to have it. As you may know, the economical situation in Lebanon is particularly bad maing any experiment impossible right now. We agree that it is not a scientific explanation, but doing science requires money and hands and Lebanon is so bad now a days that the lab is closed.  

DPPH is a widely used assay to evaluate the antioxidant capacity; however, it is a synthetic radical and does not provide an approximate information about what might happen in the human organism. Thus, the work quality should have been improved if authors had studied the antioxidant activity against reactive species with biological significance.

Conclusion should be more concise and should highlight the main findings of the work.

Thank’s for your comment. We did it.

Reviewer 2 Report

The work is well thought out and presented in an understandable manner. The topic of the article is scientifically topical as the antioxidant and anti-inflammatory potential of extracts from natural sources (Phosphocarpus tetragonolobus) has been studied. Modern diseases are on the rise and are often associated with inflammatory processes in the body. The plant world is rich in biologically active substances that have numerous pharmacological properties (antioxidant, anti-inflammatory, anticancer, etc.) without accompanying side effects, unlike synthetic drugs.

Therefore, this article likes many other papers on similar topics, contributes to the elucidation of the mechanisms of action of biologically active substances from natural sources.

I suggest to accept the paper after a minor revision.

Words ˝in vitro˝ and Latin plant names should be corrected in Italic throughout the text.

Chapter 2.4 Phytochemical investigation.

It is necessary to describe in more detail how each test was made.

If I have made it difficult to decide about further article status with my suggestion, I believe additional review of other reviewers will help. 

Author Response

Reviewer 2 :

The work is well thought out and presented in an understandable manner. The topic of the article is scientifically topical as the antioxidant and anti-inflammatory potential of extracts from natural sources (Phosphocarpus tetragonolobus) has been studied. Modern diseases are on the rise and are often associated with inflammatory processes in the body. The plant world is rich in biologically active substances that have numerous pharmacological properties (antioxidant, anti-inflammatory, anticancer, etc.) without accompanying side effects, unlike synthetic drugs.

Therefore, this article likes many other papers on similar topics, contributes to the elucidation of the mechanisms of action of biologically active substances from natural sources.

I suggest to accept the paper after a minor revision.

Words ˝in vitro˝ and Latin plant names should be corrected in Italic throughout the text.

Thank you four your comment. We modified them.

Chapter 2.4 Phytochemical investigation.

It is necessary to describe in more detail how each test was made.

Thank’s for your comment we did it.

Reviewer 3 Report

The article studies biological activities (antioxidant and anti-inflammatory) of the total extracts from Winged beam leaves. The extracts are obtained by two methods - conventional solid-liquid extraction and ultrasound assisted extraction.

The paper can be divided in two parts - production of extracts and their testing and analyses for bioactivity.

While the second part is made at high professional level, the extraction part has some shortcomings. The maceration under constant stirring is held so long time, probably unnecessary long. My personal experience with extraction of big variety of vegetables shows that usually a couple of hours are sufficient for reaching equilibrium. As authors claim, long processing may deteriorate the bioactivity. Also, 70% ethanol may not be the optimal solvent concentration. Often it is about 45-55% and has to be experimentally determined.

The authors presume the temperature influence on the yield and bioactivity is negative. This is not a universal truth. Often a higher than room temperature (but still modest) brings higher yields without reducing activity due to better solubility. Reduction of bioactivity is usually observed after 45-55oC concerning the temperature-unstable substances.

As a whole, the maceration process is carried out at non-optimal conditions, which does not allow for correct comparison of the yields obtained by the two methods. By the way, the yields of total extracts are not reported, which seems unusual. 

So, the authors have put the stress on the study of bioactivity and not on the processes of producing the extracts. I would recommend a future more detailed study on the extraction process in order to define the optimal processing parameters and determine the quantity of total extract from unity of raw material at optimal conditions. This is an important information for practical applications. As the bioactivities of extracts obtained by the two techniques are similar, the decision which method for production is to be preferred should be taken based on the quantity of dry extract obtained from unity of raw material and processing price.

Some technical remarks:

  • Introduction is somehow longer than necessary. Rows 273-292 should be placed not in Discussion, but in Introduction;
  • Many abbreviations are used. Some of them are not defined in explicit form.
  • Row 107 – inconvenient should be inconvenience;
  • Rows 125-126 state dry leaves were ground to a homogeneous fine powder using a pestle and ceramic mortar, while row 136 states dry leaves were grinded using Polymix grind mill. Which one?
  • Row 215 – presence OF saponins;
  • Row 234 – 2 should be two;
  • Row 236 – the same as 234;
  • Row 235 – Dry extract is put into solution with concentration 200 micrograms/ml. Which solvent? Probably 70% ethanol? Say it.
  • Rows 244-250 (processing description) should precede rows 236-239 (results).The same for 265-271 and 257-260;
  • Row 247 – results are should be results were (sequence of tenses);
  • Row 269 – the same as 247;
  • Row 254 – mediator’s should be mediators;
  • Row 311 – discovery, better recovery;
  • Row 320 – tetragonolobusis should be tetragonolobus is;
  • Row 325-327 – no verb in the sentence.

An important contribution of the paper is the qualitative phytochemical screening and the results for anti-oxidant and anti-inflammatory activities. The tests with the two extracts are carried out at a high professional level.

The manuscript presents the first biological and analytical study on this species. New specific information is obtained, which is useful for future studies and for possible practical applications. I recommend publishing of this article after minor revision.

Author Response

Reviewer 3 :

The article studies biological activities (antioxidant and anti-inflammatory) of the total extracts from Winged beam leaves. The extracts are obtained by two methods - conventional solid-liquid extraction and ultrasound assisted extraction.

The paper can be divided in two parts - production of extracts and their testing and analyses for bioactivity.

While the second part is made at high professional level, the extraction part has some short comings. The maceration under constant stirring is held so long time, probably unnecessary long. My personal experience with extraction of big variety of vegetables shows that usually a couple of hours are sufficient for reaching equilibrium. As authors claim, long processing may deteriorate the bioactivity. Also, 70% ethanol may not be the optimal solvent concentration. Often it is about 45-55% and has to be experimentally determined.

Thank you for your comment. The maceration at ambiante temperature does not detriorate the bioactive compounds unlike the reflux and ultrasound assisted extraction. Hijazi, A.;Bandar, H.;Rammal, H.;Hachem, A.;Saad, Z.; Badran, B. Techniques for the Extraction of Bioactive Compounds from Lebanese Urtica dioica. 2013.

Concerning the concentration of the solvent, 70% is an optimal solvent concentration according to this review that is why we use it. Zhang QW, Lin LG, Ye WC. Techniques for extraction and isolation of natural products: a comprehensive review. Chin Med. 2018;13:20. Published 2018 Apr 17. doi:10.1186/s13020-018-0177-x

The authors presume the temperature influence on the yield and bioactivity is negative. This is not a universal truth. Often a higher than room temperature (but still modest) brings higher yields without reducing activity due to better solubility. Reduction of bioactivity is usually observed after 45-55oC concerning the temperature-unstable substances.

As a whole, the maceration process is carried out at non-optimal conditions, which does not allow for correct comparison of the yields obtained by the two methods. By the way, the yields of total extracts are not reported, which seems unusual. 

So, the authors have put the stress on the study of bioactivity and not on the processes of producing the extracts. I would recommend a future more detailed study on the extraction process in order to define the optimal processing parameters and determine the quantity of total extract from unity of raw material at optimal conditions. This is an important information for practical applications. As the bioactivities of extracts obtained by the two techniques are similar, the decision which method for production is to be preferred should be taken based on the quantity of dry extract obtained from unity of raw material and processing price.

You are right but, our purpose was to study the antioxydant and anti-inflammatory properties of the biological extracts of Lebanese Psophocarpus tetragonolobus using two extraction methods maceration and ultrasound, not to compare the yield of the two technics. Also we added a description of the extraction for each bioactive compound.

Some technical remarks:

  • Introduction is somehow longer than necessary. Rows 273-292 should be placed not in Discussion, but in Introduction;
  • Many abbreviations are used. Some of them are not defined in explicit form.
  • Row 107 – inconvenient should be inconvenience;
  • Rows 125-126 state dry leaves were ground to a homogeneous fine powder using a pestle and ceramic mortar, while row 136 states dry leaves were grinded using Polymix grind mill. Which one?
  • Row 215 – presence OF saponins;
  • Row 234 – 2 should be two;
  • Row 236 – the same as 234;
  • Row 235 – Dry extract is put into solution with concentration 200 micrograms/ml. Which solvent? Probably 70% ethanol? Say it.
  • Rows 244-250 (processing description) should precede rows 236-239 (results).The same for 265-271 and 257-260;
  • Row 247 – results are should be results were (sequence of tenses);
  • Row 269 – the same as 247;
  • Row 254 – mediator’s should be mediators;
  • Row 311 – discovery, better recovery;
  • Row 320 – tetragonolobusis should be tetragonolobus is;
  • Row 325-327 – no verb in the sentence.

An important contribution of the paper is the qualitative phytochemical screening and the results for anti-oxidant and anti-inflammatory activities. The tests with the two extracts are carried out at a high professional level.

The manuscript presents the first biological and analytical study on this species. New specific information is obtained, which is useful for future studies and for possible practical applications. I recommend publishing of this article after minor revision.

We thank you for your comments that they had undoubtelly a good impact on this manuscript.

We did all the needed changes based on your comments.

Round 2

Reviewer 1 Report

I read with interest the revised version of the manuscript. The work quality was improved. Indeed, it would be interesting to have knownledge of the real chemical composition of the extracts and to perform relationships between the chemistry and the biological activities, but I understand the authors explanation about the economical situation of Lebanon. Thus, I have only one suggestion regarding Table 3, as flavanones are one classe of flavanoids, and both of them are phenols, authors should present the results of phenols, followed by flavonoids and followed by flavanones.  

Author Response

Dear Reviewers, Dear Editor,

Thank you for reviewing our manuscript entitled «Study The Antioxydant and Anti-inflammatory properties of the biological extracts of Psophocarpus tetragonolobus Using Two Extraction Methods» and for providing the opportunity to resubmit and to address the comments of the referees. Thus, please find below our response to your comments. We updated the manuscript accordingly.

Yours sincerely,

Marc Maresca

(on behalf of all authors)

Reviewer:

I read with interest the revised version of the manuscript. The work quality was improved. Indeed, it would be interesting to have knownledge of the real chemical composition of the extracts and to perform relationships between the chemistry and the biological activities, but I understand the authors explanation about the economical situation of Lebanon. Thus, I have only one suggestion regarding Table 3, as flavanones are one classe of flavanoids, and both of them are phenols, authors should present the results of phenols, followed by flavonoids and followed by flavanones.  

Answer: Thank you for your understanding and for your comments. We did what you suggest.

This manuscript is a resubmission of an earlier submission. The following is a list of the peer review reports and author responses from that submission.

Round 1

Reviewer 1 Report

Dear Authors,

Dear authors,

The responses to my comments are not reflected in the document. furthermore, the articles that you mention as justification for the qualitative determination of secondary metabolites are not part of the literature cited in the manuscript. I cannot observe any improvement in the manuscript.

Reviewer 2 Report

There are still things that need to be corrected in the manuscript.

The reference part needs many corrections. (I didn't mark whole parts) Please refer to the latest papers in this journal and revise them.

The resolution of Figure 1 looks low, so please correct it.

There are too many parts that are not indented for each paragraph.

Reviewer 3 Report

It was a pleasure for me to read the manuscript of Bassal and colleagues. The authors use the phytoextracts of P. tetragonolobus (P.t.) to evaluate its anti-inflammatory action. In particular, the authors compare two products of P.t. obtained with two extractive modalities and evaluate the protective action towards an inflammatory stimulus such as LPS in RAW264.7 cells. 

Furthermore, the authors evaluate the expression of mRNA and proteins involved in inflammatory signaling demonstrating and confirming the anti-inflammatory action of these phytoextracts.

I confirm the scientific validity of this manuscript, the authors should only correct some typos in the text. I also suggest two manuscripts that could enhance the bibliography of their manuscript:

Mastinu A et al. Protective Effects of Gynostemma pentaphyllum (var. Ginpent) against Lipopolysaccharide-Induced Inflammation and Motor Alteration in Mice. Molecules. 2021

Gupta AK et al. Artocarpus lakoocha Roxb. and Artocarpus heterophyllus Lam. Flowers: New Sources of Bioactive Compounds. Plants (Basel). 2020